# Spotlight on Accessory Proteins: RTK-RAS-MAPK Modulators as New Therapeutic Targets

**DOI:** 10.3390/biom11060895

**Published:** 2021-06-16

**Authors:** Silke Pudewell, Mohammad Reza Ahmadian

**Affiliations:** Institute of Biochemistry and Molecular Biology II, Medical Faculty, Heinrich-Heine University, 40225 Düsseldorf, Germany; sipud101@uni-duesseldorf.de

**Keywords:** adaptor proteins, anchoring proteins, docking proteins, KRAS, scaffold proteins, RAF kinase, RTK, MEK, ERK

## Abstract

The RTK-RAS-MAPK axis is one of the most extensively studied signaling cascades and is related to the development of both cancers and RASopathies. In the last 30 years, many ideas and approaches have emerged for directly targeting constituent members of this cascade, predominantly in the context of cancer treatment. These approaches are still insufficient due to undesirable drug toxicity, resistance, and low efficacy. Significant advances have been made in understanding the spatiotemporal features of the constituent members of the RTK-RAS-MAPK axis, which are linked and modulated by many accessory proteins. Given that the majority of such modulators are now emerging as attractive therapeutic targets, a very small number of accessory inhibitors have yet to be discovered.

External signals are sensed, integrated, and amplified by conserved signaling cassettes. The activation of the RTK-RAS-MAPK signaling cascade is regulated by several different extracellular signals and intracellular proteins. Growth factors activate receptor tyrosine kinases (RTKs) at the plasma membrane, which in turn activate RAS via the GDP/GTP exchange reaction. These reactions are catalyzed by guanosine exchange factors (GEFs), such as SOS1 (son of sevenless 1). GTP-bound RAS initiates RAF dimerization and induces the phosphorylation cascade towards MEK and ERK [1]. Aberrant regulation or hyperactivation of the pathway leads to cancer and a group of developmental disorders with a mild gain-of-function of the RAS-MAPK pathway, which are collectively called RASopathies [2,3].

Targeting the constituent components of the RTK-RAS-MAPK signaling cascade often leads to high toxicity and activation of backup mechanisms, lowering the treatment efficacy and increasing the burden of the therapy. In this context, we have to consider that RASopathies are caused by germline mutations, therefore, most patients are children.

Here, we highlight a group of proteins named “accessory proteins”, which are emerging as new potential therapeutic targets for the treatment of RAS-MAPK-related diseases. These proteins orchestrate the assembly and spatiotemporal localization of the constituent members of the cascade, without being part of the signaling pathway themselves [4]. Accessory proteins can be categorized into four distinct subgroups (see Figure 1): (1) anchoring proteins that bind to the membrane and other effectors (mostly kinases); (2) docking proteins that bind to receptors (e.g., RTKs and GPCRs) and more than one effector; (3) adaptor proteins that simply link two signaling components (e.g., receptor and GEF); and (4) scaffold proteins that bind two or more partners and provide a signaling platform.

As modulators of the RTK-RAS-MAPK axis, accessory proteins are multidomain proteins that bind several interaction partners and connect them as a functional unit. Accordingly, they contribute to liquid–liquid phase separation (LLPS) events, which are crucial for a directed assembly of the respective signaling machinery [5,6]. Furthermore, they can fine-tune the crosstalk between signaling pathways, increase the dwell time of proteins on the membrane, induce nanoclustering, sequester effectors, and shield them from activation, or determine the cell type specificity and subcellular localization of signaling cassettes [7]. These modulating abilities turn accessory proteins into incredibly flexible and important proteins within a very specific context. In fact, it is easy to understand why the dysregulation of accessory proteins not only leads to cancer development and cancer progression in RAS-mutant tumors but also contributes to RASopathies.

The last 30 years of research have led to significant discoveries and improvements in cancer treatment, but new therapeutic approaches are still needed, especially for cancers with KRAS mutations. Several accessory proteins have been suggested to be promising targets in RAS-mutant cancer treatment, but a very limited number of inhibitors have yet to be discovered. The major advantage of targeting modulators rather than main players is that the hyperactive signaling mechanism is attenuated and reduced to a physiological level, but not abolished, through robust inhibition. For example, the knockout of KSR in mice does not abolish ERK phosphorylation completely, furthermore, it is quite well tolerated while mouse development and is resistant against RAS-driven tumor formation [8]. This can lead to less toxicity, particularly regarding side effects (including feedback mechanisms) and decrease the burden of the treatment [9]. An example of the advantage of targeting modulators is the case of the scaffold protein SHOC2; depletion of this protein leads to a better response with MEK inhibitor treatment by interfering with the feedback mechanism towards RAF [10]. SHOC2 binds PP1 and MRAS in a holoenzyme complex, which enables RAF dimerization by dephosphorylation and the release of RAF from its autoinhibited state. The knockout of SHOC2 in mice is embryonic lethal but quite well tolerated in adult animals and human cell lines, and the knockout of SHOC2 leads to growth inhibition of RAS-mutant cell lines [10]. In addition to cancer involvement of SHOC2, mutations of this protein were also detected in Mazzanti syndrome (a RASopathy) and in prenatal-onset hypertrophic cardiomyopathy [11,12]. The mutations cause continuous membrane localization or enhanced binding of MRAS and PPP1CB, respectively, and highlight the fine-tuned signaling modulation of the scaffold protein. Another example is the anchoring protein CNK1. It localizes at the membrane via a pleckstrin homology (PH) domain and binds RAF and RAS with different C- and N-terminal domains, facilitating RAF activation and MAPK signaling [13]. The PH-domain inhibitor PHT-7.3 effectively prevents the colocalization of CNK1 with membrane-localized RAS and inhibits cell growth of KRAS-mutant cancer cell lines but not KRAS wild-type cell lines [14].

These two examples highlight the importance of a tightly controlled spatiotemporal organization of signaling components through accessory proteins and are just small insights into the large group of these modulators [4]. More research can pave the way for new therapeutic strategies involving single and cotreatment approaches that directly target the specific scaffold, adaptor, docking, or anchoring function of accessory proteins.

## Figures and Tables

**Figure 1 biomolecules-11-00895-f001:**
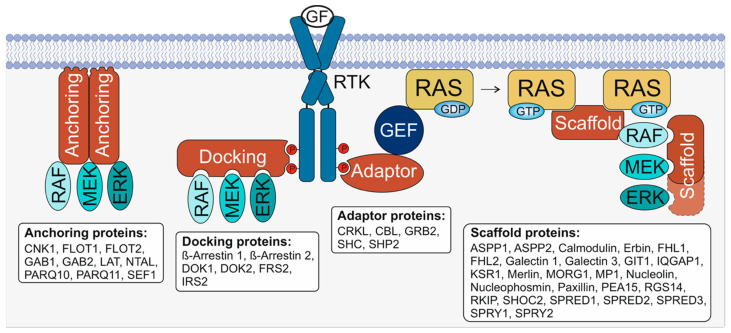
Accessory proteins can be divided into at least four subgroups: anchoring, docking, adaptor, and scaffold proteins. All groups include several different binders of the RTK-RAS-MAPK signaling pathway, such as RAF, MEK, and/or ERK, and often share common domains. Anchoring proteins include membrane-associated domains, such as pleckstrin homology (PH) and transmembrane (TM) domains or a posttranslational modification, (e.g., myristoylation in FLOT2) to determine a special subcellular localization and increase the dwell time of the proteins at the membrane. Docking proteins connect receptors with downstream effectors and feature receptor binding domains, such as PTBs (phosphotyrosine binding domains). FRS2, for example, interacts with the FGFR (fibroblast growth factor receptor) and links activated RTKs (receptor tyrosine kinases) with adaptor proteins such as GRB2 and SHC, and with other effectors, such as SOS1. Adaptor proteins can simply connect two proteins and often exhibit SH (SRC homology) 2 and SH3 domains. GRB2 is a well-known adaptor protein of RTKs and SOS1 but can also, as mentioned before, bind to other accessory proteins and RTKs and fine-tune the signaling machinery and the cross-talk of different pathways. Scaffold proteins can simultaneously bind several signaling components. They can contribute to clustering events by oligomerization via special domains or intrinsically disordered regions (such as galectin 1 and 3), recruit proteins to the site of action (such as SPRED1), or determine RAS signaling at a specific subcellular localization (such as MP1, which binds MEK1 and ERK1 on late endosomes). Scaffold proteins are also able to bind other accessory proteins and allow tight control of the RAS-MAPK pathway.

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
