# Peer review of "Spotlight on Accessory Proteins: RTK-RAS-MAPK Modulators as New Therapeutic Targets"

_biomolecules, 2021, doi:10.3390/biom11060895_

Round 1

Reviewer 1 Report

The commentary is well-written and comes in a timely manner, as we continue to understand the roles of accessory proteins in signaling through the MAPK pathway. It is short and to the point. Comments/question:

  • The 14-3-3 proteins are conspicuously missing from Figure 1, as it plays a key role in Raf kinase regulation. Please add or explain why it is excluded.
  • The following two sentences would benefit from references.

“The big advantage of targeting modulators rather than main players is that the hyperactive signaling mechanism is attenuated and reduced to a physiological level but not abolished (ref). This can lead to less toxicity, particularly regarding the side effects and the burden of the treatment (ref)”.

The importance of these references is that these statements are at the heart of motivation for targeting the accessory proteins.

  • The PH-domain inhibitor PHT-7.3 is mentioned. Are there any other examples of inhibitors/drugs targeting adapter proteins? If so, a compilation of those that are available and their target accessory proteins would be interesting and strengthen the commentary.

Author Response

Authors’ response to the reviewers’ comments:
Reviewer #1:
The commentary is well-written and comes in a timely manner, as we continue to understand the roles of accessory proteins in signaling through the MAPK pathway. It is short and to the point. Comments/question:

1) The 14-3-3 proteins are conspicuously missing from Figure 1, as it plays a key role in Raf kinase regulation. Please add or explain why it is excluded.
Authors’ response: We thank the reviewer for raising this point.
We have excluded 14-3-3 proteins from our lists of accessory proteins for the RTK-RAS-MAPK axis for the following reasons: (i) 14-3-3 proteins bind to distinct phosphoserine and phosphothreonine sites of signaling proteins, and controls signal transduction both in a negative and positive manner; so, there is no clear directionality. (ii) They operate in different ways, scaffolding, stabilizing a distinct conformation, and masking and occluding functional sites. (iii) They bind to an astonishingly large number of binding partners (more than one hundred). Thus, loss or gain of 14-3-3 functions have many different consequences, and 14-3-3 proteins are as such suboptimal drug targets.

2) The following two sentences would benefit from references.
“The big advantage of targeting modulators rather than main players is that the hyperactive signaling mechanism is attenuated and reduced to a physiological level but not abolished (ref). This can lead to less toxicity, particularly regarding the side effects and the burden of the treatment (ref)”. The importance of these references is that these statements are at the heart of motivation for targeting the accessory proteins.
Authors’ response: We thank the reviewer for critical suggestions.
The attention towards accessory proteins rises but there are not too many specific inhibitors discovered yet. Therefore, we included an example in this section to support the main statement of this commentary. For KSR1 it could be shown that the bi-allelic knockout does not abolish MEK and ERK phosphorylation completely and the knockout is quite well tolerated by mice but are resistant to RAS driven tumor formation (Nguyen et al. Mol. Cell. Biol., 2002). The patient’s treatment burden increases by prolonged and multiple therapies which could be due to drug resistance after a feedback loop in the signaling mechanism. The example of the SHOC2 knockout which interferes with the feedback mechanism of MEK inhibition in RAS mutated cancers is already included (Jones et al. Nat. Commun, 2019).

3) The PH-domain inhibitor PHT-7.3 is mentioned. Are there any other examples of inhibitors/drugs targeting adapter proteins? If so, a compilation of those that are available and their target accessory proteins would be interesting and strengthen the commentary.
Authors’ response: We thank the reviewer for bringing up this point. Yes, there are also a few drugs for adaptor proteins, e.g., SHP099 against SHP2 and liposome-incorporated antisense oligodeoxynucleotides towards GRB2. SHP099 is inhibiting the adaptor function, as well as the phosphatase activity of SHP2 and is used in RAS-mutated cancers (Xie, J. et al. J. Med. Chem, 2017). The anti-GRB2 miDNA targets and knocks down GRB2 mRNA and inhibits its translation. The drug was used in leukemia with the Philadelphia-chromosome (Tari et al. Int. J. Oncol, 2007). However, both drugs do not only interfere with the accessory function of these proteins as the PH-domain inhibitor of CNK1 does. We discussed more inhibitors in our latest perspective in communications biology (Pudewell et al.  2021) and will refer to this paper as it would exceed the frame of a commentary.

Reviewer 2 Report

The manuscript by Pudewell and Ahmadian is a brief commentary emphasizing the potential role of accessory proteins of the RTK-RAS-MAPK pathway in cancer and RASopathies. The manuscript is well written but gives only a glimpse of what could be their relevance to treat these diseases. Given that the topic has not received the necessary attention in the scientific literature, this commentary will be important to raise more awareness. In my opinion, this thematic area even deserves a full review.

The only comment I have is that cancers are never mutated only in KRAS4B but both KRAS isoforms. This needs to be corrected.

Author Response

Reviewer #2:
The manuscript by Pudewell and Ahmadian is a brief commentary emphasizing the potential role of accessory proteins of the RTK-RAS-MAPK pathway in cancer and RASopathies. The manuscript is well written but gives only a glimpse of what could be their relevance to treat these diseases. Given that the topic has not received the necessary attention in the scientific literature, this commentary will be important to raise more awareness. In my opinion, this thematic area even deserves a full review.
The only comment I have is that cancers are never mutated only in KRAS4B but both KRAS isoforms. This needs to be corrected.
Authors’ response: We thank the reviewer for the encouraging comments.
We changed KRAS4B in the manuscript to KRAS mutated cancers.

Reviewer 3 Report

It is very concise and written clearly. I do not have any major comment. 

I just have the blew minor suggestion:

One of the main mediator of the RTKs is the Hippo signalling pathway. I think it is important to discuss about it and cite two below papers:

A gain-of-functional screen identifies the Hippo pathway as a central mediator of receptor tyrosine kinases during tumorigenesis, Oncogene volume 39pages334–355(2020)

Author Response

Reviewer #3:
It is very concise and written clearly.